# The Impact of a Multicomponent Platform Intervention on the Daily Lives of Older Adults

**DOI:** 10.3390/healthcare11243102

**Published:** 2023-12-05

**Authors:** Vera Stara, Margherita Rampioni, Adrian Alexandru Moșoi, Dominic M. Kristaly, Sorin-Aurel Moraru, Lucia Paciaroni, Susy Paolini, Alessandra Raccichini, Elisa Felici, Giacomo Cucchieri, Luca Antognoli, Alessandra Millevolte, Marina Antici, Mirko di Rosa

**Affiliations:** 1Model of Care and New Technologies, IRCCS INRCA-National Institute of Health and Science on Aging, Via Santa Margherita 5, 60124 Ancona, Italy; v.stara@inrca.it (V.S.); e.felici@inrca.it (E.F.); g.cucchieri@inrca.it (G.C.); l.antognoli@inrca.it (L.A.); 2Department of Psychology, Education and Teacher Training, Transilvania University of Brasov, B-dul Eroilor 29, 500036 Brașov, Romania; adrian.mosoi@unitbv.ro; 3Department of Automatics and Information Technology, Transilvania University of Brasov, B-dul Eroilor 29, 500036 Brașov, Romania; dominic.kristaly@unitbv.ro (D.M.K.); smoraru@unitbv.ro (S.-A.M.); 4Neurology Department, IRCCS INRCA-National Institute of Health and Science on Aging, Via della Monta-gnola 81, 60100 Ancona, Italy; l.paciaroni@inrca.it (L.P.); s.paolini@inrca.it (S.P.); a.raccichini@inrca.it (A.R.); 5Laboratorio delle Idee, Via G.B. Miliani 36, 60044 Fabriano, Italy; alessandramillevolte@gmail.com (A.M.); marina.antici@labidee.com (M.A.); 6Unit of Geriatric Pharmacoepidemiology and Biostatistics, IRCCS INRCA-National Institute of Health and Science on Aging, 60124 Ancona, Italy; m.dirosa@inrca.it

**Keywords:** gerontechnology, active and healthy aging, older adults, technology-based intervention, independent living, usability, acceptance

## Abstract

Gerontechnology is an interdisciplinary field of research involving gerontology and technology in order to help older adults identify and slow down the effects of age-related physical and cognitive decline. It has enormous potential to allow individuals to remain in their own homes and improve their quality of life. This study aims to assess the impact of a multicomponent platform, consisting of an ambient sensor, wearable devices, and a cloud application, as an intervention in terms of usability and acceptance as primary outcomes and well-being, quality of life, and self-efficacy as secondary outcomes in a sample of 25 older adults aged over 65 after 21 days of non-supervised usage at home. This research involved the use of a mixed-methods approach, in which both qualitative and quantitative data were collected in three different measurements. Overall, the participants shared good engagement with the integrated platform. The system achieved positive results in terms of both usability and acceptance, especially the smartwatch. The state of complete well-being slightly improved over the period, whereas self-efficacy remained stable. This study demonstrates the ability of target users to use technology independently in their home environment: it strengthens the idea that this technology is ready for mainstream use and offers food for thought for developers who create products for the aging population.

## 1. Introduction

Gerontechnology is an interdisciplinary field of research and application involving gerontology, the scientific study of aging which examines the biological, psychological, and sociological factors associated with the aging process [1] and technology that can help older adults identify, prevent, and slow down the effects of age-related physical and cognitive declines [2]. Its areas of competence are technology for independent people (e.g., work and self-fulfillment), assistive technology (e.g., housing and health), and technology for communication and leisure. Therefore, gerontechnology has enormous potential to ensure better care and an improved sense of security and quality of life for older adults [3], but it also clashes with three different types of barriers, such as decreased cognitive functions, accessibility, and a lack of knowledge about technology [4,5]. In fact, although gerontechnology is really supportive of daily living, over the last few decades, several studies have investigated the numerous reasons why older individuals are less inclined to use technology [6].

Chung et al. [7] observed that for aging-in-place technology to be truly adopted by older adults, devices should address seniors’ values, self-perceptions, and ethical issues at the intersection of aging, technology, and home environment [8]. Older adults highlight that the main deciding factors on whether to use or accept technologies are that these technologies should allow them to remain in their own homes and age in place, improve their quality of life, and be perceived as useful [9], especially in the tracking of health [10].

On one hand, older adults tend to associate the adoption of new technologies with a lack of confidence in their ability to understand or access them, resulting in a sense of frustration and inadequacy [11]. The concept of usability is therefore linked to the ease of use of the devices and the possibility of taking advantage of short training, perhaps with the help of caregivers. Since lower self-efficacy and higher computer anxiety predict lower use of technology, the learning perspective in the development and deployment of new technologies should play a significant role so as to facilitate access by older users [12].

On the other hand, privacy is a critical factor affecting older adults’ acceptance of smart home technology [9]. Seniors want to feel that they have the power to control their lives and surroundings for as long as possible, and not to be controlled. They want to decide whether and how to use the technology, when it is turned on or off, where it is placed, and with whom to share the collected data. Stakeholders often underline the value of empowerment and self-management to promote well-being [13]. Autonomy for smart home technology means “the assistive technology developed for elderly care must not interfere with the will of the person it is designed to care for” [14]. Older adults want to be perceived as strong, capable, and independent individuals, and this desire stems from their wish to not be perceived as a burden to family, friends, or society, in general [8]. Any device that projects negative aging stereotypes is likely to be rejected by older adults, even if the device is helpful [15]. A recent mapping review on studies of successful aging via assistive information and communication technologies [6] mapped a heterogeneous scenario of research gaps where mixed methods research and the cooperation among clinical, technical, societal, and research areas might help in reaching the effectiveness of solutions to improve the quality of life of older adults [6].

In our view, it could be stimulating for designers and stakeholders starting from the description of individual cases: in fact, when home monitoring technology for aging in place is used appropriately, it can improve independence and quality of life, maintain the health and well-being of older adults, facilitate their engagement in everyday tasks, and support caregivers and healthcare professionals [16,17]. In this work, a combined approach integrating ambient sensors and wearable monitoring devices is associated with a network platform that gives access to caregivers, researchers, and third-party services to address the needs of older adults. For all of these reasons, this manuscript aims to assess two research questions: (1) to what extent usability and acceptance can impact a multicomponent platform intervention and (2) to what extent a multicomponent platform intervention improves well-being, quality of life, and self-efficacy in a sample of 25 older adults aged over 65 with a Mini-Mental State Evaluation (MMSE) [18] score between 21 and 27 after 21 days of non-supervised usage at home.

Therefore, this paper first describes the study design and the participants’ interaction with the SAVE platform, then discusses the results obtained in terms of usability, acceptance, well-being, and self-efficacy, and finally offers food for thought for developers that aim to create products that meet the end users’ needs [19], and could also be a source of inspiration for larger studies [6].

## 2. Materials and Methods

### 2.1. Study Design

This study involves the use of the SAVE platform for a total of 21 consecutive days. We recommended that the participants use the system every day and we checked the time of interactions through the SAVE cloud application. The research was managed by qualified personnel, who ensured both the supervision of the tests by specialists and the detailed measurement of the first interaction between the users and the prototype system. This personal guarantee also facilitated the training on the use of the technology. The study involved the use of a mixed-methods approach, in which both qualitative (open questions) and quantitative (standardized tests) data were collected in three different measurements: (1) at time 0 before the start of the experimentation (T0); (2) at time 1, after 10 days, i.e., at the midterm of the trial (T1); and (3) at time 2, after 21 days, i.e., at the end of the trial (T2). The log data of the usage of the SAVE system were continuously stored over the 21-day test period.

The study was approved by the Ethical Committee of Istituto di Ricerca e Cura a Carattere Scientifico, Istituto Nazionale Ricovero e Cura Anziani (IRCCS INRCA) and then registered on the platform ClinicalTrials.gov with the registration number NCT05626556.

### 2.2. Participants

The study involved 25 primary users who met the inclusion and exclusion criteria described in Table 1.

The participants had an average age of 81.2 years and were represented by women at 64% and men at 36%. The majority of them (48%) were widowed, 44% were married, and 8% were single. A total of 44% of the seniors had a primary education level, 28% had a secondary education level, and 28% had a tertiary education level (university or further education). Approximately 44% of users live in the city, 40% live in a village, and 16% live in a county town. Approximately 92% of the participants are retired, whilst 8% are still working. Among the latter, more than half of users (52%) work full-time or are self-employed. Approximately 12% of participants report having family duties/obligations, e.g., taking care of grandchildren. The majority of the sample (52%) carries out leisure activities (e.g., dance, reading, or chess), and only 32% of users regularly exercise (e.g., walking or swimming).

The average MMSE score (25.6 ± 1.9) was between 21 and 27, a range that was predefined in the inclusion criteria. The average FAC score (4.8 ± 0.5) shows that the users have the ability to achieve autonomy in walking and the average BI score (19.1 ± 2.8) highlights that they have a fairly independent functional status, that is, the ability to perform normal daily activities required to meet basic needs and fulfill usual roles. Table 2 shows the users’ characteristics.

### 2.3. The Intervention

The concept of the SAVE system is that of a multi-component platform intervention based on multiple smart-home and wearable sensors streamed directly to a cloud-based platform, where algorithms detect any behavioral and physiological information about older adults’ well-being and security in their habitat. This platform was developed under the framework of a European-funded project (EU Grant Agreement AAL-CP-2018-5-149), named SAfety of elderly people and Vicinity Ensuring—SAVE—by following a user-driven approach [20].

As shown in Figure 1, the SAVE solution consists of:A kit of sensors (Aqara Home) composed of 1 control unit, 1 smoke sensor (kitchen), 2 flood sensors (kitchen and bathroom), 1 contact sensor (entrance door), and 2 presence sensors;A smartwatch (Samsung Galaxy Watch3) with the SAVE software pre-installed (Version 1.0; Website: SAVE Web App (saveaal.eu). The smartwatch included a wide range of sensors, which could be used to measure physical activity (the number of steps and their frequency and the speed of movement), obtain some basic biological signals (e.g., pulse beat), provide an SOS service (call to the caregiver), and detect possible falls. This information was then used to assess the well-being of users;A SAVE Sensors Adapter.

All tools were connected to the SAVE cloud application.

The SAVE system was installed in end users’ homes, by positioning all of the kits in the appropriate rooms and by offering relevant training to the end-users for using all of the different devices. The sensors were located in places where they do not affect the daily activity of the users, with them being easy to move according to preferences and small enough to blend into the background. Thus, at the end of the installation, the users had the Aqara Home System in their homes, a Samsung smartwatch, and a SAVE Sensors Adapter, all of which were connected to a router with unlimited Internet access.

After installing the system, a brief instruction training session on the use and purpose of these devices was performed. This ensured that the end users were encouraged to use them, and we performed some tests with them:Heart rate testing in association with the frequency indicated on the smartwatch;Testing the emergency system by pressing the power button 3 times;Testing of the flood and door sensors by visualizing the values received by the SAVE cloud app through the SAVE web app;Calling a friend/relative from their smartwatch.

### 2.4. The Outcomes

The primary outcomes were:

Usability is understood as “the extent to which a product can be used by certain users to achieve certain goals with effectiveness, efficiency and satisfaction in a given context of use”. This result was measured through the SUS scale [21] and the UEQ-S questionnaire [22].

Acceptance is the degree to which users come to accept and use a form of technology. This result was measured through the QUEST 2.0 scale [23].

The secondary outcomes were:

Well-being is understood as “a state of complete physical, mental and social well-being, and not simply as the absence of disease”. This result was measured through the WHO-5 Index [24] and the EQ-5D-5L questionnaire [25].

Self-efficacy is understood as the set of beliefs we have about our ability to complete a certain task. This result was measured through the short version of the GSE self-efficacy scale [26].

### 2.5. Study Setting

The Istituto di Ricerca e Cura a Carattere Scientifico and Istituto Nazionale Ricovero e Cura Anziani (the IRCCS INRCA is located in the city of Ancona in Italy and the Laboratorio delle Idee is located in the town of Fabriano) managed the enrollment of users and data collection.

### 2.6. Data Collection

Data collection was started in August 2022 and finished in June 2023. Three different data collection tools were developed. Their details are reported in Table 3.

### 2.7. Data Analysis

The users’ characteristics were described by reporting the mean and standard deviation for continuous variables and the absolute number and percentage for categorical variables. The distribution of SF-12 domains, Who-5, GSE, and EQ-5D at baseline (T0) and at the end of the trial (T2) and the distribution of SUS and QUEST scores at the midterm of the trial (T1) and T2 are displayed together with the median and 95% confidence interval (95%CI). The User Experience Questionnaire (UEQ-S) was analyzed by using the Compare Scale Means for UEQ-S tool by Schrepp [30]. All other analyses were performed by using GraphPad Prism version 8.0.0 for Windows (GraphPad Software, San Diego, CA, USA).

## 3. Results

All data reported in the following subparagraphs are shown in Appendix A.

### 3.1. Smartwatch Usage

Figure 2a illustrates the relationship between the daily minutes of smartwatch usage and the average number of steps taken each day by Italian users. The majority of users do not exceed an average of 4000 steps per day, regardless of the duration of smartwatch usage. The average total daily steps taken by users is 2724 steps. The mean daily usage of the smartwatch corresponds to 595 min (9.9 h). Considering that the smartwatch requires daily charging (recommended for at least two hours, ideally overnight), the acceptability is high, with an average daily usage of 9.9 h. Figure 2b displays the total number of days of smartwatch usage for each user. The red line represents the average days of usage by all users, which is 16.5 days.

### 3.2. Usability and Acceptance

All participants successfully completed the SUS. The SUS is scored out of 100, with a higher score indicating greater perceived usability. As shown in Figure 3a, the system received a mean score of 63.8 ± 22.8 at T1 and 65.3 ± 20.4 at T2. These scores were compared and interpreted considering the acceptable average value of 68 (SD 12.5), which was determined for a variety of products and tools, including websites and technologies, provided by Sauro and Lewis [31], after the analysis of more than 5000 user scores encompassing almost 500 studies.

As shown in Figure 3b, the mean QUEST score for the smartwatch varied from 31.2 ± 5.7 at T1 to 32.5 ± 4.8 at T2, and, for the kit of sensors, from 35.4 ± 4.9 at T1 to 36.0 ± 4.1 at T2, indicating a level of ‘quite satisfied’ overall with the system.

As regards user experience, in Figure 4, different results were reported. The perspicuity (i.e., the ease of becoming familiar with the product), as well as the efficiency (i.e., the ability to solve tasks without unnecessary effort), increased over time. On the contrary, the dependability (the feeling of being in control of the interaction) and the stimulation (i.e., the motivation to use the product) remained stable, whereas the attractiveness (i.e., the overall impression of the product) and the novelty, that is, the power of the system to catch the interest of the users, decreased in the last period of usage.

### 3.3. Well-Being and Self-Efficacy

As reported in Figure 5a, without any change in the perceived health status monitored through the SF-12, the state of complete physical, mental, and social well-being, as measured through the WHO-5 Index and the EQ-5D-5L questionnaires, slightly improved over the period, with it ranging from 68.0 ± 15.7 to 69.4 ± 13.7 in the case of WHO-5 (Figure 5b) and from 71.2 ± 13.5 to 74.0 ± 12.7 in the case of EQ-5D-5L (Figure 5c). Self-efficacy as the set of beliefs about the ability to complete a certain task was measured through the short version of the GSE self-efficacy scale, and it remained stable (Figure 5d).

## 4. Discussion

The purpose of this study was to assess the impact of a multicomponent platform intervention in terms of usability and acceptance as the primary outcomes and well-being, quality of life, and self-efficacy as the secondary outcomes in a sample of 25 older adults aged over 65 with a Mini-Mental State Evaluation (MMSE) [18] score between 21 and 27 after 21 days of non-supervised usage at home. Even if this study recruited users in the early stages of cognitive decline, they all used devices independently and were still able to manage routine changes, such as introducing novel technological tools into their daily lives [32]. Indeed, the participants involved in this study demonstrated a positive approach, they all provided useful feedback to facilitate the understanding of the data gathered, and only one participant withdrew from the trial. It is well known that for people living with cognitive decline, all novelties can produce distress. In this study, users benefited from general training on the correct use of the different system components (i.e., the smartwatch and kit of sensors) and were given two dedicated phone numbers to call for technical assistance or clarity of doubts related to the proper use of the system. As reported in the results, becoming familiar with such technology seems to be a good property to assure good usability, user experience, and acceptance outcomes. This finding is in line with other studies in which older adults appeared to be compliant with the use of innovative solutions [33]. In a recent scoping review [34], the key to matching the n needs and requirements of older adult users to ensure accessibility and usability is considered a priority in the design phase of any device. An in-depth understanding of users, their needs, and the contexts in which the systems are used are significant information to consider [2]. Effectively, in the reported study, this activity was carried out from the beginning in order to ensure the user-centered design [20].

The aging population is interested and competent in using these health technologies, especially wearables and portables [35], with the use of such technologies reaching an average of 829.85 min/day [36]. In our study, users reached an average of 595 min/day. Compliance may depend on the ease of use and the perceived benefits obtained by the use of technologies. This is true also with technology that is not at a high technology readiness level (TRL) as in the case of the SAVE platform, and this can demonstrate that the technology is ready for mainstream use as reported in other studies [15]. In particular, those technologies that keep track of and measure physiological parameters can be widely used and spread in the silver market. Unfortunately, as reported in the state-of-the-art literature, a difficult barrier to overcome is the poor experience of the majority of older adults in using this technology [2,34] which can be overcome by supporting users with dedicated assistance in case of need.

Literature research indicates that technology-based interventions that are appropriately designed can bring many benefits to older people, such as increasing independence, maximizing physical and mental health, and improving their quality of life [37], and the data collected through this study partially confirms this idea. This is probably due to a minimum time of exposition, 21 days, which did not engage the older adults for a long time to appreciate such benefits. Indeed, the methodological approach such as the design of the study presents challenges [6].

### Limitations of the Study and Future Directions

The first limitation of this study concerns the readiness level of the technology used which may have impacted the results. Even if the TRL was low, it was in line with a recent systematic review [9] that lists a huge number of studies in the field that are still in the development and testing phase (TRL5) or in the demonstration phase, pilot phase, and prototypical system validation (TRL6). Another limitation is related to the small sample size enrollment, gender disparities, and the specific Italian national context and culture which could be seen as a bias that does not allow for the generalization of the results. Also, the limited time of usage of the system is a limitation; in the specific case of this study, this was due to the time and cost boundaries of a funded research project. By the way, longer trials are needed to measure changes in user experience and familiarity with the system. Furthermore, sharing the strengths and weaknesses of research is fundamental for building common knowledge from previous studies. New technologies are reaching great utility in helping to maintain health and self-care more and more, independence, and autonomy to maintain or increase individuals’ quality of life [2]. Technology-based interventions seem to focus on a single domain of intervention (physical, cognitive, mental, emotional, etc.) rather than approach the multinational promotion of health [34]. Focus on such a multidimensional approach could increase the necessary availability of evidence to move the actual debate one step forward. This study holds significant implications, especially for those individuals invested in the development of technologies designed to provide support and monitoring for seniors facing cognitive decline. It markedly emphasizes the prospect of progressing towards an approach that encompasses the integration of sensors and multi-component strategies, all while adhering to stringent criteria regarding acceptability and usability. This suggests that future initiatives in the field could benefit from the approach outlined in the study, which focuses on harmonizing advanced technical functionalities with attention to practicality and acceptance among elderly users.

## 5. Conclusions

This study aimed to evaluate the usability and acceptance of a multi-component platform intervention by 25 users living with cognitive decline. It demonstrated the ability of target users to use technology independently in their home environment. Overall, the participants shared good engagement with the integrated system. Despite limitations, this study strengthens the idea that gerotechnology is ready for mainstream use and it would offer food for thought for developers who design products for the aging population or could be used as a source of inspiration for larger studies. These kinds of devices have enormous potential to ensure better care and an improved sense of security and quality of life for older adults [3]. Indeed, in the last decade, digital healthcare technologies have rapidly grown in diagnostics, therapy, assistance, epidemiology, and other sectors [38], but, as demonstrated by this study, when such tools match the older population market, different barriers must be considered. In this study, we focused on usability and acceptance, and we also mentioned the lack of knowledge on how to use and benefit from technology [4,5]. Even if the demographic trends will solve the e-health literacy issue and maybe legal and safety concerns will also be solved [39], the largest studies and strongest cooperation among sectors (i.e., technology, health and care, research, policy, and social sectors) still remain vital to proof the real impact of technology in the healthcare field.

## Figures and Tables

**Figure 1 healthcare-11-03102-f001:**
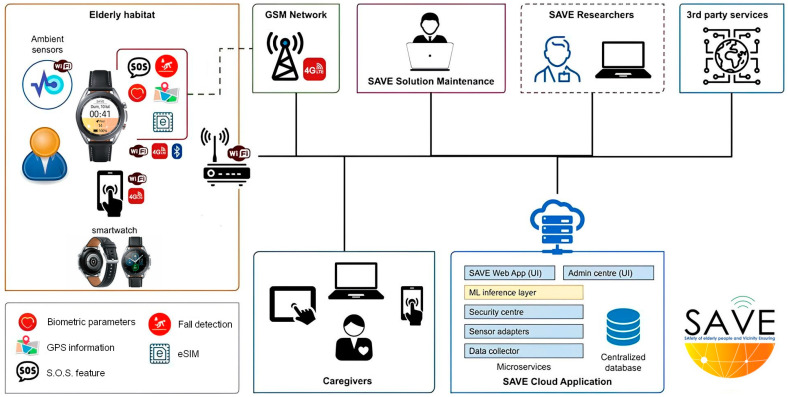
The SAVE multicomponent platform.

**Figure 2 healthcare-11-03102-f002:**
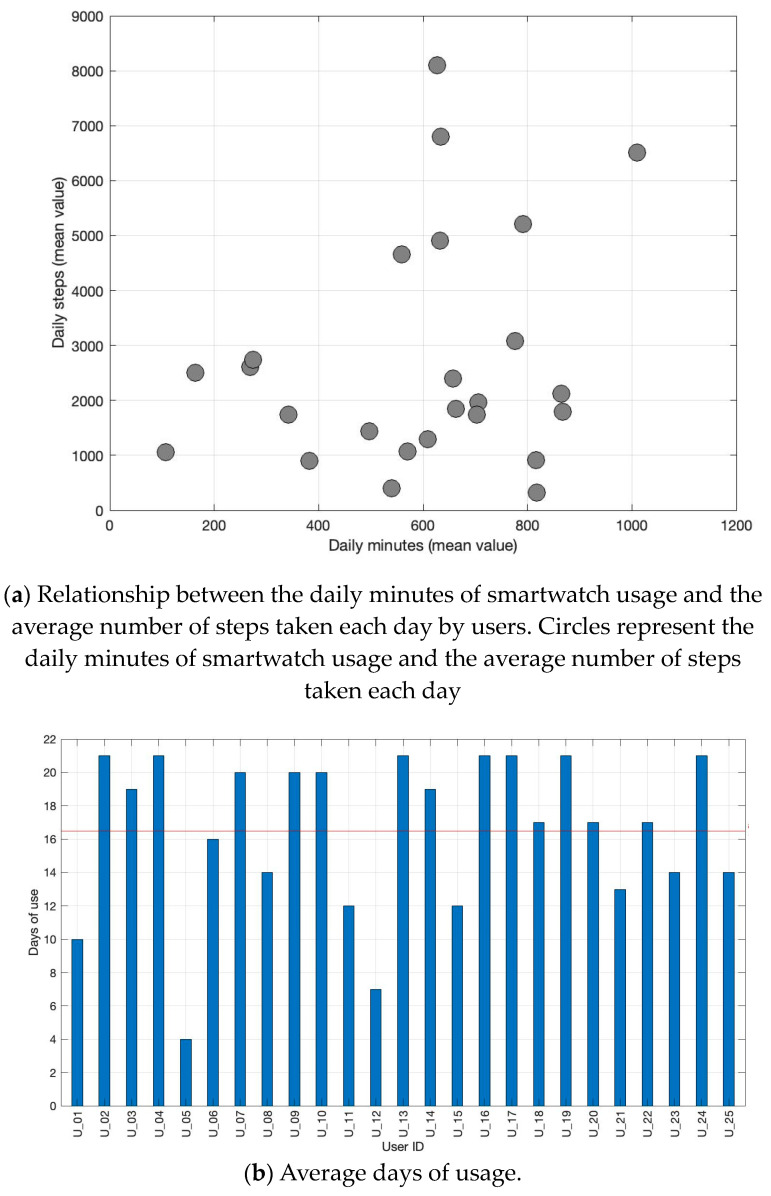
Correlation between the mean daily minutes of smartwatch usage and the mean daily number of steps taken by each Italian user (**a**) and the number of days of smartwatch usage for each user (**b**). The red line represents the average days of usage.

**Figure 3 healthcare-11-03102-f003:**
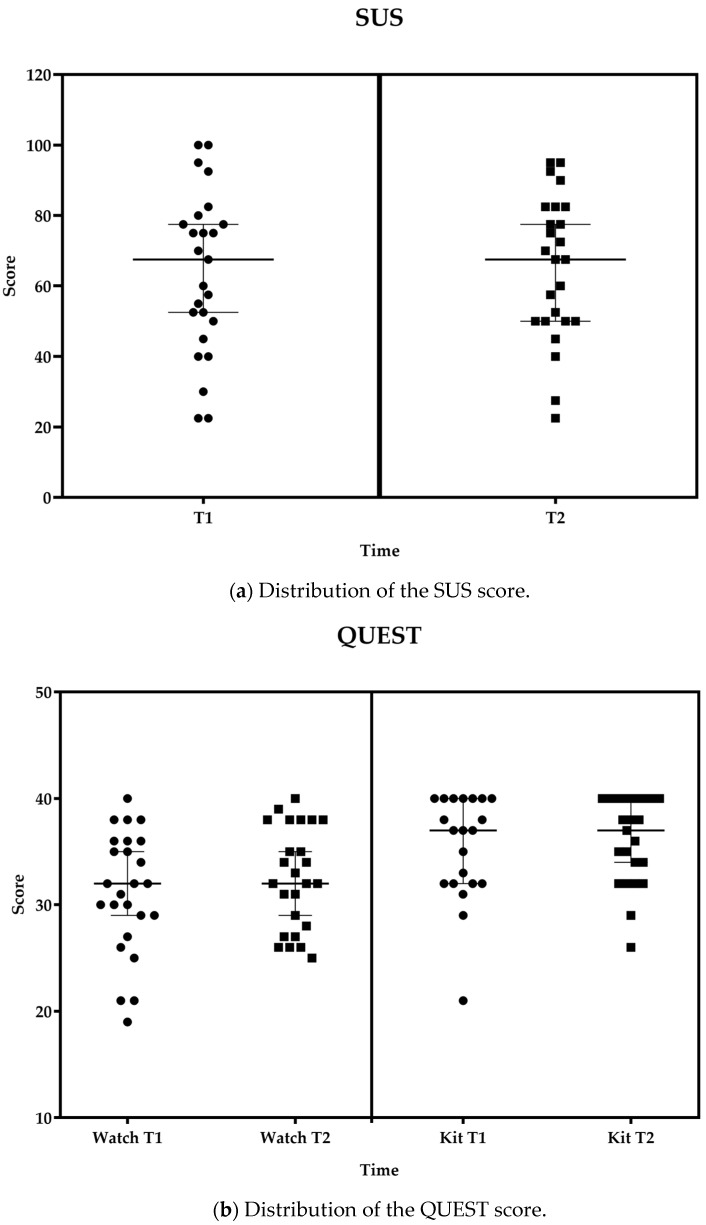
Distribution of SUS (**a**) and QUEST (**b**) scores for the smartwatch and kit of sensors at the midterm of the trial (T1) and T2 displayed together with the median and 95% confidence interval (95%CI). Boxes represent values at T2; Circles represent values at T0 or T1 as appropriated.

**Figure 4 healthcare-11-03102-f004:**
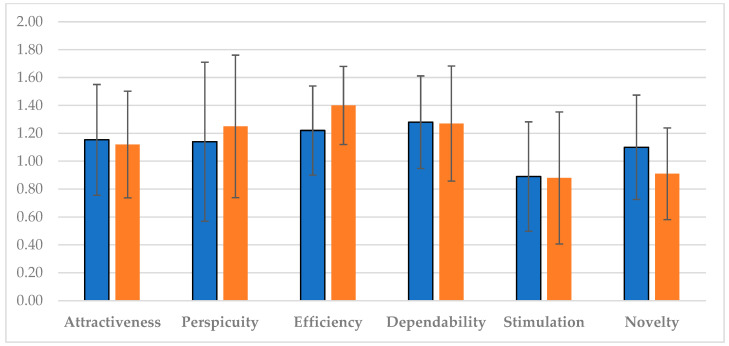
The user experience scale means and the corresponding 5% confidence intervals at T1 (blue color) and T2 (orange color).

**Figure 5 healthcare-11-03102-f005:**
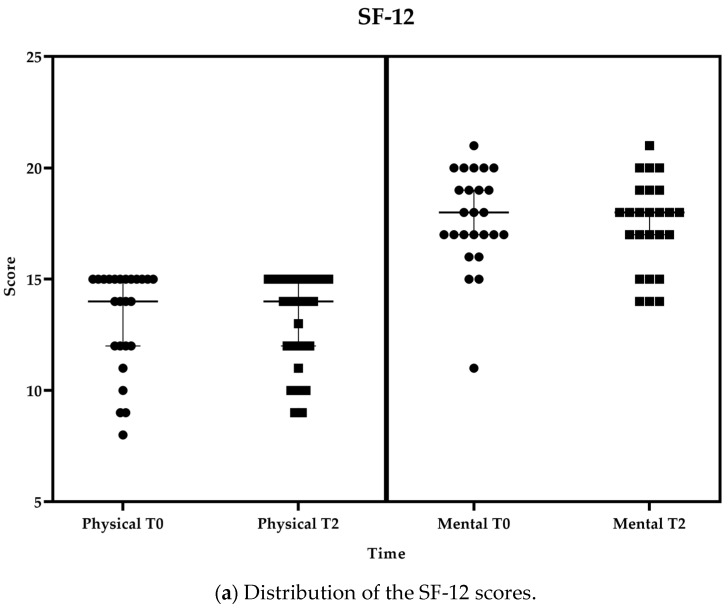
Distribution of SF-12 (**a**), WHO-5 (**b**), EQ-5D-5L (**c**), and GSE (**d**) at T1 and T2 displayed together with the median and 95% confidence interval (95%CI). Boxes represent values at T2; Circles represent values at T0 or T1 as appropriated.

**Table 1 healthcare-11-03102-t001:** Eligibility criteria for primary users.

End User Type	Inclusion Criteria	Exclusion Criteria
	- Age ≥ 65 years - Mini-Mental State Evaluation (MMSE) score between 21 and 27 - Healthy or mild-to-moderate chronic illness or musculoskeletal disease - Feel physically fit to participate in the study (assessment with FAC and Barthel index): sufficiently capable of moving, able to maintain and change their position, manipulate and move objects, move in their place of residence, experience the surrounding environment, and move by means of transport - Live alone or with a spouse - Interest in the project - Able to perform the tasks suggested by the caregiver - Able to use a smartphone and smartwatch - The presence of a caregiver	- Age < 65 years - MMSE < 21, subjects diagnosed with dementia or MMSE ≥ 28 or ≤20 - Participants suffering from severe chronic disease (e.g., symptomatic cardiovascular or respiratory disease, myocardial infarction, or stroke in the last 6 months or the presence of significant visual and/or auditory impairment, severe metabolic dysfunction, and oncological pathologies) or severe disability - Participants who are carriers of cardiac pacemakers or implantable defibrillators - The presence of conditions that make it difficult to use a smart device (e.g., moderate/severe dementia, aphasia, etc.) - A person placed under guardianship - Nickel allergy

**Table 2 healthcare-11-03102-t002:** Users’ characteristics.

	All Users (N = 25)
Age, mean ± sd	81.2 ± 6.8
Gender, n (%)	
Male	9 (36.0%)
Female	16 (64.0%)
Marital status, n (%)	
Married	11 (44.0%)
Single	2 (8.0%)
Widowed	12 (48.0%)
Education, n (%)	
Primary	11 (44.0%)
Secondary	7 (28.0%)
Tertiary	7 (28.0%)
Settlement, n (%)	
County town	4 (16.0%)
City	11 (44.0%)
Village	10 (40.0%)
Retired, n (%)	23 (92%)
Job, n (%)	
Full-time/Self-employed	13 (52.0%)
None	11 (44.0%)
NA	1 (4.0%)
Family duties, n (%)	3 (12.0%)
Leisure activities, n (%)	13 (52.0%)
Physical exercise, n (%)	8 (32.0%)
MMSE, mean ± sd	25.6 ± 1.9
FAC, mean ± sd	4.8 ± 0.5
BI, mean ± sd	19.1 ± 2.8

**Table 3 healthcare-11-03102-t003:** Dimensions, scales, and timing of data collection.

Type of End Users	Dimensions	Scales	Time		
			**T0**	**T1**	**T2**
	Health and Wellness Condition	Mini-Mental State Examination (MMSE) [6]	**X**		
Primary users		Functional Ambulation Category (FAC) [27]	**X**		
		Barthel Index (BI) [28]	**X**		
		SF-12v2™ Health Survey [29]	**X**	**X**	**X**
		Five Well-Being (WHO-5) Index [24]	**X**	**X**	**X**
		EuroQol–5 Dimension–5 Level (EQ-5D-5L) [25]	**X**	**X**	**X**
	Self-efficacy	General Self-Efficacy Scale (GSE) [26]	**X**	**X**	**X**
	Usability and Acceptance	System Usability Scale (SUS) [21]		**X**	**X**
		User Experience Questionnaire (UEQ-S) [22]		**X**	**X**
		Quebec User Evaluation of Satisfaction with Assistive Technology (QUEST 2.0) [23]		**X**	**X**
	Privacy and Stigma	Open questions	**X**	**X**	**X**

## Data Availability

The data presented in this study are available in the article itself.

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
