# Peer review of "The Impact of a Multicomponent Platform Intervention on the Daily Lives of Older Adults"

_healthcare, 2023, doi:10.3390/healthcare11243102_

Round 1

Reviewer 1 Report

Comments and Suggestions for Authors

This work may be of interest to readers and is generally well written; However, before its publication, I recommend that the following recommendations be followed:

At the end of the introduction, better describe the research gap and add at least 2 research questions that are addressed by this work.

Finish the introduction section with a paragraph that explains how the rest of the document is distributed by sections.

The paragraph above figure 2 mentions figure 1 a and b, but below it is figure 2. Correct this. Additionally, this figure mentions a correlation, however this can be confusing, because the calculations of this index are not seen and information is not given about the type of relationship, whether positive or negative, that is found. Check if the word correlation is appropriate and also review this concept in statistics.

Check the consecutive in the numbering of the figures.

Author Response

This work may be of interest to readers and is generally well written; However, before its publication, I recommend that the following recommendations be followed:

At the end of the introduction, better describe the research gap and add at least 2 research questions that are addressed by this work.

Answer: thank you for this important suggestion. We added two research questions

Finish the introduction section with a paragraph that explains how the rest of the document is distributed by sections.

Answer: done

The paragraph above figure 2 mentions figure 1 a and b, but below it is figure 2. Correct this. Additionally, this figure mentions a correlation, however this can be confusing, because the calculations of this index are not seen and information is not given about the type of relationship, whether positive or negative, that is found. Check if the word correlation is appropriate and also review this concept in statistics.

Answer: done

Check the consecutive in the numbering of the figures.

Answer: done

We really thank you for this review.

Reviewer 2 Report

Comments and Suggestions for Authors

Dear Authors,

The present research article, entitled “The impact of a multicomponent platform intervention in the daily life of older adults”, aims to assess the impact of a multicomponent platform 25 intervention in terms of usability and acceptance as primary outcomes, and well-being, quality of 26 life and self-efficacy as secondary outcomes in a sample of 25 older adults aged over 65 after 21 days 27 of non-supervised usage at home.

In general, I believe that the topic and approach of this article is timely and of interest to the readers of Healthcare. However, I believe that some issues should be included to improve the quality of the manuscript.

Abstract:

·         I think it would be useful to specify a little and concisely what the intervention is based on using the multicomponent platform. When " smartwatch" is mentioned in the results, it generates confusion in the readers.

·         How many days a week did they perform it? Was it controlled that everyone did it to the same extent?

Introduction

·         I think it would be useful to describe briefly and concisely what this multicomponent platform is all about. There is a very good justification as to why technology may be needed for older adults but not what type and/or what characteristics the technology that this article focuses on has. A justification of related work on the platform they present would facilitate understanding.

·         Please add the study hypothesis, which was missing in the manuscript

Methods

·         I think it is not a good idea to include informed consent as an inclusion/exclusion criterion. It is taken for granted that the study will be conducted only with people who have given consent. A statement, at a general level, about that is sufficient.

·         I think it would be more appropriate to start the method with the participants and the design than with the intervention. Generally, the subsections follow an order. I recommend that you review the journal's directions on this.

Results

·         The fundamental characteristics of the participants should be described in the method.

·         Figure a and b should be specified.

·         What is the basis for the statement " As the system used in this research was a system prototype 226 and not a product ready for the market, it reached a positive result even if the score is 227 below the acceptable average score of 68"? It is not understood why the justification that it was considered acceptable because it is a prototype.

·         Who facilitated the training on the use of the technology?

Discussion

·         I believe that more emphasis should be placed on comparison with previous literature.

·         Future impact could be maximized.

 Best regards.

Author Response

The present research article, entitled “The impact of a multicomponent platform intervention in the daily life of older adults”, aims to assess the impact of a multicomponent platform 25 intervention in terms of usability and acceptance as primary outcomes, and well-being, quality of 26 life and self-efficacy as secondary outcomes in a sample of 25 older adults aged over 65 after 21 days 27 of non-supervised usage at home.

In general, I believe that the topic and approach of this article is timely and of interest to the readers of Healthcare. However, I believe that some issues should be included to improve the quality of the manuscript.

Abstract:

  • I think it would be useful to specify a little and concisely what the intervention is based on using the multicomponent platform. When " smartwatch" is mentioned in the results, it generates confusion in the readers.

Answer: done

  • How many days a week did they perform it? Was it controlled that everyone did it to the same extent?

Answer: we specified this point on 2.1 Study deisgn: “The study involves the use of the SAVE platform for a total of 21 consecutive days. We recommended to the participants to use the system every day and we check the time of interactions through the SAVE cloud application”

Introduction

  • I think it would be useful to describe briefly and concisely what this multicomponent platform is all about. There is a very good justification as to why technology may be needed for older adults but not what type and/or what characteristics the technology that this article focuses on has. A justification of related work on the platform they present would facilitate understanding.

Answer: done

  • Please add the study hypothesis, which was missing in the manuscript

Answer: done

Methods

  • I think it is not a good idea to include informed consent as an inclusion/exclusion criterion. It is taken for granted that the study will be conducted only with people who have given consent. A statement, at a general level, about that is sufficient.

Answer: done

  • I think it would be more appropriate to start the method with the participants and the design than with the intervention. Generally, the subsections follow an order. I recommend that you review the journal's directions on this.

Answer: done

Results

  • The fundamental characteristics of the participants should be described in the method.

Answer: done

  • Figure a and b should be specified.

Answer: done

  • What is the basis for the statement " As the system used in this research was a system prototype 226 and not a product ready for the market, it reached a positive result even if the score is 227 below the acceptable average score of 68"? It is not understood why the justification that it was considered acceptable because it is a prototype.

Answer: thank you for this comment. We prefer to delete the sentence to avoid confusion.

  • Who facilitated the training on the use of the technology?

Answer: we specified this point on 2.1 Study deisgn: “The research was managed by qualified personnel, who ensure both the supervision of the tests by specialists and the detailed measurement of the first interaction between the users and the prototype system. This personal guaranteed also facilitated the training on the use of the technology”

Discussion

  • I believe that more emphasis should be placed on comparison with previous literature.
  • Future impact could be maximized.

Answers: thank you for theses suggestion. We better describe the comparison.with previous literature using recent systematic and scoping reviews and we added the “future directions”

We really thank you for this review.

Round 2

Reviewer 2 Report

Comments and Suggestions for Authors

I think your modifications to the text are very accurate.